# A Novel Method for Filled/Unfilled Grain Classification Based on Structured Light Imaging and Improved PointNet++

**DOI:** 10.3390/s23146331

**Published:** 2023-07-12

**Authors:** Shihao Huang, Zhihao Lu, Yuxuan Shi, Jiale Dong, Lin Hu, Wanneng Yang, Chenglong Huang

**Affiliations:** 1College of Engineering, Huazhong Agricultural University, Wuhan 430070, China; hshhzau@163.com (S.H.); 19108410785@163.com (Z.L.); 15342769488@163.com (Y.S.); dongjl2022@163.com (J.D.); 15971177620@163.com (L.H.); 2Shenzhen Institute of Nutrition and Health, Huazhong Agricultural University, Wuhan 430070, China; 3Shenzhen Branch, Guangdong Laboratory for Lingnan Modern Agriculture, Genome Analysis Laboratory of the Ministry of Agriculture, Agricultural Genomics Institute at Shenzhen, Chinese Academy of Agricultural Sciences, Shenzhen 518000, China; 4National Key Laboratory of Crop Genetic Improvement, National Center of Plant Gene Research (Wuhan), Huazhong Agricultural University, Wuhan 430070, China; ywn@mail.hzau.edu.cn

**Keywords:** 3D structured light, point cloud segmentation, data enhancement, deep learning, normal vector, grain classification

## Abstract

China is the largest producer and consumer of rice, and the classification of filled/unfilled rice grains is of great significance for rice breeding and genetic analysis. The traditional method for filled/unfilled rice grain identification was generally manual, which had the disadvantages of low efficiency, poor repeatability, and low precision. In this study, we have proposed a novel method for filled/unfilled grain classification based on structured light imaging and Improved PointNet++. Firstly, the 3D point cloud data of rice grains were obtained by structured light imaging. And then the specified processing algorithms were developed for the single grain segmentation, and data enhancement with normal vector. Finally, the PointNet++ network was improved by adding an additional Set Abstraction layer and combining the maximum pooling of normal vectors to realize filled/unfilled rice grain point cloud classification. To verify the model performance, the Improved PointNet++ was compared with six machine learning methods, PointNet and PointConv. The results showed that the optimal machine learning model is XGboost, with a classification accuracy of 91.99%, while the classification accuracy of Improved PointNet++ was 98.50% outperforming the PointNet 93.75% and PointConv 92.25%. In conclusion, this study has demonstrated a novel and effective method for filled/unfilled grain recognition.

## 1. Introduction

Rice is one of the most important crops in the world, and more than 65% of the Chinese people take rice as their staple food. So, the rice yield was of great significance to the food security of the world [1]. With the explosive growth in population and the shortage of water resources and land, we face severe challenges in agricultural production [2,3,4]. The rice yield measurement was necessary for rice breeding and genetic research, while the grain setting rate of rice was a significant determinant of rice yield [5]. The number of filled grains per panicle was directly related to crop yield [6], so it was very important to distinguish between filled/unfilled rice grains, which was an indispensable step in the rice yield measurement. This study aimed to find an efficient and accurate method for filled/unfilled rice grain identification.

In order to distinguish filled/unfilled grains, the water sedimentation or air separation method was usually adopted to separate the filled grains from the unfilled grains, which suffered from low efficiency, poor repeatability, and low accuracy [7,8]. Water sedimentation and air separation method utilize the internal cavity of the unfilled grains to achieve the separation of the filled/unfilled grain by adjusting external forces. To improve the identification of filled/unfilled grains, Duan et al. proposed a method based on visible light and X-ray imaging for fast discrimination and counting of filled/unfilled rice grains [9]. However, the X-ray imaging method was expensive and had radiation risks. Kumar et al. developed an automated imaging system based on thermal images to separate and calculate filled rice grains [10]. But this method required heating the grains, which had the weakness of complicated operation, and low efficiency. Traditional research methods focus on two-dimensional images, which provided limited information. However, filled/unfilled rice grains would be better discriminated in three-dimensional space. Point clouds obtained through binocular stereo vision, motion structure, and space carving were relatively sparse, and better results may be achieved with higher-quality point clouds [11,12,13,14]. When using the commonly used binocular vision Iterative reconstruction technology, camera parameter calibration must be required for each shot, and the reconstructed model needed a large amount of data processing to eliminate noise, which would take a long time. At the same time, the obtained 3D point cloud was sparse, which was not suitable for rice grain point cloud reconstruction [15]. 3D Structured light imaging could greatly reduce the impact of environmental noise when obtaining high-density rice grain point clouds with texture information through feature stitching algorithm and texture mapping. Qin et al. have tried to classify the filled/unfilled rice grains based on structured light imaging and traditional machine learning methods, with a classification accuracy of 90.18% [16]. Although this method proved the feasibility of a three-dimensional point cloud to achieve rice grains classification, the accuracy needs further improvement. Therefore, it was of great importance to developing a high-efficiency, low-cost, and reliable method for filled/unfilled rice grain identification.

With the rapid development of computer vision and artificial intelligence technology, various deep learning networks have been proposed to solve a range of agricultural challenges. Jin et al. used the Resnet network to classify different rice varieties to achieve a classification accuracy of 86.08% [17]. Qin et al. proposed a deep learning method based on improved PointNet to classify 8 different varieties of rice grains with an accuracy of 89.4% [18]. Since the quality of point cloud data directly affected the results of point cloud processing, data enhancement was generally adopted to improve the point cloud. Kankare et al., used the point cloud down-sampling method to increase the processing speed and keep the accuracy steady [19]. Hubner et al., proposed a method of indoor point cloud classification based on a normal vector to improve the classification accuracy by 10% [20]. Therefore, the combination of the deep learning method and 3D point cloud would provide a new feasible way for higher accuracy of filled rice grain classification. Moreover, the data enhancement based on up-sampling and normal vector fusion would provide a new way for the rice grain point clouds improvement.

This study aimed to propose a novel method for filled/unfilled grain classification based on structured light imaging and Improved PointNet++. The structured light imaging was adopted to obtain three-dimensional point cloud data of rice grains. And PointNet++ was improved to achieve accurate classification of filled/unfilled rice grain. This study would provide a new approach for the effective identification of filled/unfilled rice grains, which was of great significance for rice breeding and genetic research.

## 2. Materials and Methods

### 2.1. Experimental Materials and Devices

In this study, the experimental materials were selected from the japonica and indica rice varieties Zhonghua 11 and 9311 respectively, in which the rice grains were randomly selected for the datasets. The filled and unfilled grains were taken in 50/50 proportion, which was to ensure sample balance during the filled/unfilled grain classification. Each rice variety comprised of 500 filled rice grains and 500 unfilled rice grains, resulting in a total of 1000 rice grains. The rice grains were randomly selected, while the filled and unfilled rice grains were identified by air separation.

In this study, a 3D structured light scanner based on white LED raster scanning technology (Reeyee Pro, Wiiboox, Wuhan, China) was utilized, and the parameters of the scanner were presented in Table 1. The scanner combined the advantages of structured light and binocular stereo vision, while the point precision could reach 0.05 mm, making it suitable for high-precision scanning of small workpieces, plastic products, and medical equipment. The main equipment consisted of a projector, binocular camera, texture camera, and modulated light source. The scanning area was about 210 × 150 mm, and the effective imaging distance was within the range of 290–490 mm.

The schematic diagram of structured light imaging was depicted in Figure 1, which was able to obtain dense point cloud data of objects based on the triangulation principle. When scanning an object with structured light, deformation fringes were observed due to the fringe phase change, which was used to reconstruct the three-dimensional information of the object. Moreover, the color information was also acquired by the texture mapping, which provided important data for filled/unfilled grain classification.

### 2.2. 3D Point Cloud Data Acquisition and Processing

The rice grain point clouds acquisition was conducted as shown in Figure 2. Before the point cloud acquisition, the checkerboard calibration was conducted to achieve position correction of the structured light scanner, and the white balance calibration was performed to correct the color information. Generally, there were two kinds of schemes to fix rice grains during the scanning by structured light, including laying the grain flat on a platform and fixing the grain through a seed holder [21]. The former method was highly efficient but yielded low precision due to incomplete scanning of particles. Conversely, the latter could obtain a complete single-point cloud with high accuracy, but could only scan one particle at a time. In this study, rice grains were mounted vertically on the sample plate by adhesive tape, which would provide complete point clouds and achieve multi grains imaging. Due to the limitation of the scanning area, the placement area of rice grains was restricted to the center 100 × 100 mm of the sample plate. Additionally, the distance between adjacent rice grains was set to 20 mm to avoid the shielding between rice grains, and the grain placement strategy was 5 × 5, with five grains in each row and five rows in each column. The 3D point cloud data of grains were obtained by scanning 8 times through a rotating table with an interval of 45 degrees.

The schematic diagram of grain segmentation and data enhancement was conducted as shown in Figure 3. After obtaining the rice grains point cloud data (Figure 3a), data preprocessing was conducted to achieve data enhancement. The preprocessing of the original point cloud mainly included point cloud coordinate transformation and point cloud sampling. The coordinate transformation would be shown as follows: Firstly, the covariance matrix $M_{T}$ was calculated based on principal component analysis (PCA) [22] as shown in Formula (1). Then, the original coordinate $T_{0}$ was moved to the centroid point A of the point cloud, and the new coordinate $T_{A}$ was generated according to Formula (2). After that, the random sampling consensus algorithm [23] (RANSAC) was used to separate the sample plane (Figure 3(b1)) from rice grains (Figure 3(b2)). Finally, the region growing algorithm [24] was used to identify the single particle point cloud based on curvature and normal angle (Figure 3c).
(1)$$M_{T}={\left\{e_{1},e_{2},e_{3}\right\}}^{T}$$
(2)$$T_{A}=M_{T}\times (T_{0}-A)$$
where $e_{1}$, $e_{2}$, and $e_{3}$ are the three-unit eigenvectors of the covariance matrix $M_{T}$; $T_{0}$ is the coordinates of the original point cloud; $T_{A}$ is the new coordinate after coordinate transformation; A is the point cloud centroid coordinate.

Two types of data enhancement were performed on 3D grain point clouds by moving least squares (MLS) [25,26] in our study. Firstly, in order to enquire more useful point cloud information, the up-sampling algorithm was applied to increase the point clouds density (Figure 3d), in which each grain point was sampled with uniform random distribution to maintain a constant point density. Secondly, since the difference between filled/unfilled rice grains was the curvature of the cavity in the belly of the grain, the normal vector was acquired based on the MLS method (Figure 3e) [27], which was an important feature for filled/unfilled rice grain identification. And the principle of MLS method to calculate a normal vector was as follows: finding any point $p_{i}${x,y,z} in a certain field; obtaining a plane α: $a_{0}x+a_{1}y+a_{2}z+b=0$, in which the sum S of the squares of the distances from $p_{i}$ to the plane was minimized by the Formula (3). The [$a_{0},a_{1},a_{2}$] that minimized S was the normal vector of $point\;p_{i}$.
(3)$$S=\sum_{i=1}^{n}\frac{{\left(a_{0}x_{i}+a_{1}y_{i}+a_{2}z_{i}+b\right)}^{2}}{a_{0}^{2}+a_{1}^{2}+a_{2}^{2}}$$
where [$a_{0},a_{1},a_{2}$] is the normal vector of the plane; [$x_{i},y_{i},z_{i}$] is the coordinates of point $p_{i}$; b indicates the offset of the plane from the origin.

### 2.3. 3D Grain Shape Traits Extraction

To explore the advantages of deep learning methods for filled/unfilled grain classification, this study has adopted six machine learning algorithms as comparison, including Decision Tree (DT) [28], Random Forest (RF) [29], Support Vector Machine (SVM) [30], Naive Bayes (NB) [31], Back Propagation Neural Network (BPNN) [32], and Extreme Gradient Boosting (XGBoost) [33]. The hyper parameters of all machine learning models were tuned using the learning curve and grid search techniques [34], and the accuracy of each model was verified using the 5-fold cross-validation method. 2000 grains data were randomly divided into 5 sets, and the average results of 5 trials were taken as the model accuracy. The 11 3D grain traits were extracted for the filled/unfilled grain classification, including grain length, width and height, volume surface area, projected area and projected perimeter, which was depicted in Table 2. Based on 3D point clouds, the schematic diagram of grain parameter extraction was shown in Figure 3.

The extraction of grain L, W and H were conducted as Figure 4a. Firstly, principal component analysis was used to rotate the segmented single grain point cloud for coordinate transformation, and then the axis-aligned bounding box [35] was converted into an oriented bounding box (OBB) [36]. Secondly, the maximum and minimum values of the transformed single grain point cloud, namely $x_{max}$, $x_{min}$, $y_{max}$, $y_{min}$, $z_{max}$, and $z_{min}$, were extracted based on the point coordinates. Finally, the *L*, *W*, and *H* of the grain were calculated according to Equations (4)–(6).
(4)$$L=x_{max}-x_{min}$$
(5)$$W=y_{max}-y_{min}$$
(6)$$H=z_{max}-z_{min}$$
where L, W and H are the length, width and thickness of the rice grains.

The extraction of grain volume and surface was performed as Figure 4b,c. Firstly, the convex pentahedron was constructed by triangular mesh and central plane projection algorithms. And then, the sum of the corresponding convex pentahedron volumes of all triangular meshes was calculated as the volume (*V*) of the rice grains. Secondly, the greedy projection triangulation algorithm [37] was used to establish the triangular mesh model of the rice grains point cloud. And then the mesh boundary edge reconstruction algorithm was used to fill the holes, while the edge length of the triangle was calculated from three vertices coordinates of the triangle. Finally, the grain surface (*S*) was calculated as the sum of all triangular surfaces according to Helen’s formula, which was shown as Formulas (7) and (8).
(7)$$S_{0}=\sum_{i=1}^{k}s_{i}$$
(8)$$s_{i}=\sqrt{p_{i}(p_{i}-a_{i})(p_{i}-b_{i})(p_{i}-c_{i})}$$
where $S_{0}$ is the surface area of the grain; k is the total number of triangles; $s_{i}$ is the area of the ith triangle (mm^2^); $p_{i}$ is half the perimeter of the triangle (mm); $a_{i}$, $b_{i}$ and $c_{i}$ represent the length of each side of the triangle (mm).

The projected perimeter and area of rice grain were extracted as Figure 4d. Firstly, the rice grain point cloud was projected onto the planes x = 0, y = 0, and z = 0, respectively. And then, the greedy projection triangulation algorithm was adopted to calculate the projected area and perimeter based on the triangular mesh.

### 2.4. Improved PointNet++ for Filled/Unfilled Grains Identification

The PointNet++ network was first proposed by Qi, Charles R in 2017, which was optimized on PointNet networks [38]. As for the point cloud analysis, PointNet considered the orientation splicing processing of global features, but ignored the local features of point cloud [39]. The main idea of Pointnet ++ was to continue the local partitioning of the point cloud and acieration of local features and then carried out the local division of point cloud data and feature extraction to realize hierarchical feature extraction operation. In addition, PointNet++ continuously aggregates to abstract advanced global semantic features, to decrease the influence of non-uniformity point cloud. Therefore, this study has adopted PointNet++ as the backbone network for filled/unfilled grain classification.

The Improved PointNet++ network structure was shown in Figure 5, in which the additional Set Abstraction (SA) layer, and max-pooling layer were added. Since the rice grain was small, the additional SA layer was adopted to enhance the feature extraction ability for tiny objects, the details of which was shown in Figure 6. Moreover, the maximum pooling layer was performed to optimize the disordered point clouds and redundant features. The Improved PointNet++ network was carried out including sampling, grouping, and feature extraction. The multi-scale sampling (MSG) module was depicted in Figure 7. The 1024-point clouds output from the previous sampling step were sampled with three sampling radius of 0.1, 0.2 and 0.4 mm. 512-point clouds and 256-point clouds were obtained after the first and the second sampling, respectively. In order to acquire the same number of point clouds, the point clouds in the group that were more than the given value would be removed, and if the point clouds were less than the given value, the first 16-point clouds closest to the center would be added. Due to the non-uniformity of point clouds, the features obtained in the dense area of point clouds would not be generalized to the sparse area of point clouds. The MSG sampling layer was of great importance to accurately grasp the sampling scale while point cloud density changed.

### 2.5. Environment and Model Evaluation

System environment: CPU Intel Core i7 11700K, GPU NVIDIA GeForce RTX 3060 with 12 GB of video memory. The development environment included Windows 10, Visual Studio 2015, PCL 1.8.1, Python 3.8.13, and PyCharm 2019.3.3. The library used for machine learning utilized the SkLearn 0.23.2, and the deep learning model was trained on PyTorch 1.8.0 framework. ReeyeePro V2.6.1.0 software was installed in the 3D structured light scanning device to acquire rice grains point cloud data.

The datasets were randomly divided into training and testing sets with the ratio 4:1. The batch size, learning rate, number of input point clouds, and epoch number were set as 8, 0.0001, 1024, and 60. The grain point cloud classification dataset was labeled as ModelNet40_normal_resampled format [40]. The accuracy and $F_{1}$ score were adopted as the model evaluation indicators, which were calculated as Formulas (9) and (10). In order to explore the efficiency of different improved methods for identifying filled/unfilled rice grains, the inference time of 100 rice grains was calculated as Formula (11)
(9)$$Accuracy=\frac{TP+TN}{TP+FP+TN+FN}$$
(10)$$F_{1}=\frac{2TP}{2TP+FP+FN}$$
(11)$$efficiency=\sum_{i=0}^{n}T_{i}$$
where TP, FP, TN, and FN are the number of true positives, false positives, true negatives, and false negatives, respectively; $T_{i}$ indicate the testing time of rice grain.

## 3. Results

### 3.1. Classification Results Comparison for Different Methods

The grain classification results based on different deep-learning models were shown in Table 3. Without data augmentation, the results indicated that the accuracy of PointNet, PointConv, PointNet++, and Improved PointNet++ was 87.00%, 88.25%, 91.75% and 92.96%, while the time cost of PointNet, PointConv, PointNet++, and Improved PointNet++ was 7.65 s, 12.44 s, 9.75 s and 12.75 s for 100 rice grains classification. The results proved that the PointNet++ model outperformed the PointNet and PointConv model for the grain classification, and the Improved PointNet++ would increase the accuracy by 1.21%. Due to the additional feature extraction layer and maximum pooling method, the Improved PointNet++ had the best performance for filled/unfilled grain classification.

Because the up-samping and normal vector fusion could increase point cloud density and enrich point cloud features, the data augmentation would have a remarkably positive impact on the classification results. With data augmentation, the results showed that the accuracy of PointNet, PointConv, PointNet++, and Improved PointNet++ was 93.75%, 92.25%, 97.25%, and 98.5%, while the time cost of PointNet, PointConv, PointNet++, and Improved PointNet++ was 7.98 s, 14.43 s, 13.62 s and 16.29 s for 100 rice grains classification. The results demonstrated that the data augmentation would dramatically increase the classification accuracy by over 4.00%, but only decrease the efficiency by an average of 2.43 s. The data augmentation would increase the classification accuracy by 5.54% for the Improved PointNet++, which proved to be an effective method for filled/unfilled grain classification improvement.

In order to evaluate the performance of the up-sampling and normal vector fusion method, the parameter ablation experiment was conducted based on the Improved PointNet++ model, which was shown in Table 4. The results showed that the classification accuracy with up-samping was 94.50%, which had been promoted by 1.54%. The classification accuracy with normal vector fusion was 97.75%, which had been enhanced by 4.79%. The results demonstrated that the data augmentation methods were able to significantly enhance point cloud density and expand point cloud features, compared with the raw data. As for the classification accuracy improvement, the normal vector fusion method outperformed the up-sampling method with approximately three times accuracy improvement.

### 3.2. Classification Results of Filled/Unfilled Rice Grains Based on Improved PointNet++

To explore the classification results of filled/unfilled rice grains, 200 point cloud data of rice grains were randomly selected in the testing set, and predicted by Improved PointNet++, in which the data augmentation was applied by up-sampling and normal vector fusion method. The classification result for the confusion matrix of filled/unfilled rice grains, was shown in Figure 8. The results showed that the misjudged filled/unfilled grains number of the PointNet++ model was 28 (Figure 8a), while the Improved PointNet++ model significantly reduced the misjudged number by 9 (Figure 8a), which proved that the model optimization would decrease the misjudgment of filled/unfilled grain. The results also indicated that the Improved PointNet++ model with data augmentation would remarkably decrease the misjudgment number to 7 (Figure 8c), which proved that the up-sampling and normal vector fusion could effectively improve the filled/unfilled rice grains classification. In order to verify model generalization ability, 100 grains of Fengliangyou 4 and Guangliangyouxiang 66 were tested based on the Improved PointNet++ model, and the classification accuracy was 95.83% and 98.00%, respectively.

In order to display the classification results, the rice varieties of Zhonghua 11 and 9311 with 25 rice grains were selected for the visual prediction based on the data augmentation processing and Improved PointNet++ model, which was shown in Figure 9. In the prediction box, the green and blue rice grain indicated the misjudged filled and unfilled grain. The results proved that most of the rice grains could be classified correctly. The results showed that the filled grain might be identified as unfilled grain for Zhonghua 11, and the unfilled grain might be classified as filled grain for 9311, which was related to the rice varieties.

### 3.3. Classification Results of Machine Learning Methods

In order to explore the advantages of the Improved PointNet++ model, six machine learning algorithms including DT, RF, SVM, NB, BPNN, and XGBoost were adopted for filled/unfilled rice grain classification. The hyper parameters were decided by learning curves and grid search methods, and a 5-fold cross-validation method was adopted to verify the accuracy of each algorithm [41], in which the initial learning rate was set to 0.0003. And the classification results of each algorithm were shown in Table 5. The results indicated the classification accuracy of the DT, RF, SVM, NB, BPNN, and XGBoost models were 89.94%, 90.08%, 90.96%, 88.96%, 91.06% and 91.99% respectively. The results proved that XGBoost outperformed the other machine learning methods. However, the machine learning algorithm need to manually extract the grain traits, which was time-consuming and subjective, and the results demonstrated that the Improved PointNet++ model had better performance than the XGBoost model with a 6.51% improvement.

## 4. Conclusions

The filled/unfilled rice grain discrimination was important for rice breeding and genetic analysis. However, the traditional method including water sedimentation or air separation had the disadvantages of low efficiency, poor repeatability, and low precision. In this study, a novel method for filled/unfilled grain classification was proposed based on structured light imaging and Improved PointNet++, in which data augmentation was adopted to achieve accurate classification. In conclusion, a novel and effective solution for high-precision filled/unfilled rice grains recognition was demonstrated, and the main conclusions of the study were drawn as follows.

(1)The additional feature extraction layer and maximum pooling were adopted to improve the PointNet++, which would enhance the ability for filled/unfilled grain feature extraction and reduce the disorder of grain point clouds, and the results showed that the Improved PointNet++ would increase the filled/unfilled grain classification accuracy by 1.21%.(2)The data augmentation techniques including up-samping and normal vector fusion were adopted to increase point cloud density and enrich point cloud features, which was proven to be of great significance for the filled/unfilled grain classification improvement. And the results proved that the data augmentation could promote the classification accuracy by 5.54% for the Improved PointNet++.(3)11 3D grain traits were extracted for the filled/unfilled grain classification, and six machine learning algorithms were adopted for comparisons. The results showed that the Improved PointNet++ with data augmentation outperformed the best machine learning algorithm XGBoost by 6.51%, which proved that the deep learning model had more powerful capabilities for feature extraction and filled/unfilled grain classification.

## Figures and Tables

**Figure 1 sensors-23-06331-f001:**
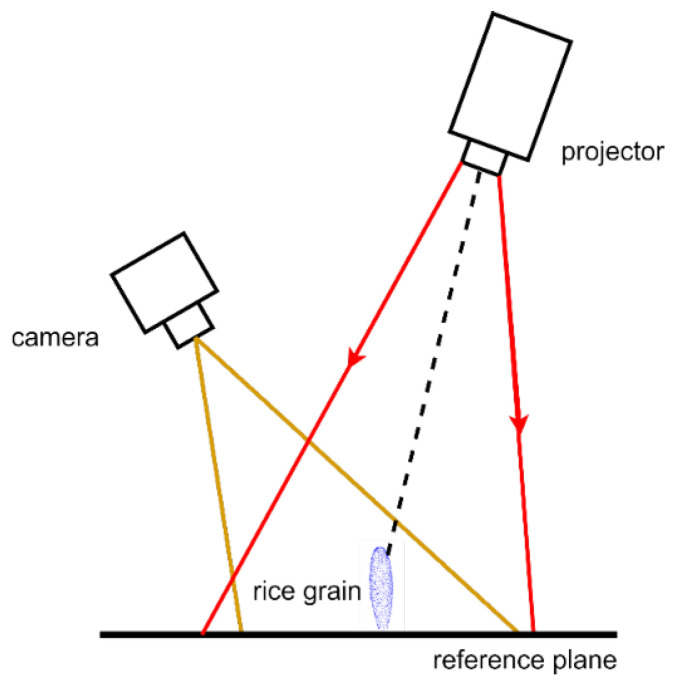
Schematic diagram of rice grain scanning based on structured light imaging.

**Figure 2 sensors-23-06331-f002:**
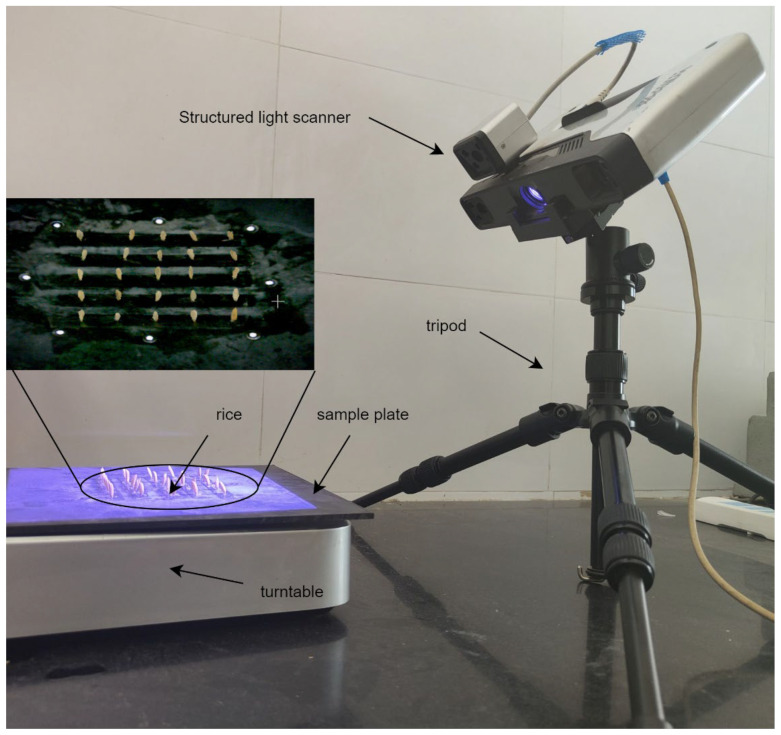
Rice grain point cloud acquisition based on structured light imaging.

**Figure 3 sensors-23-06331-f003:**
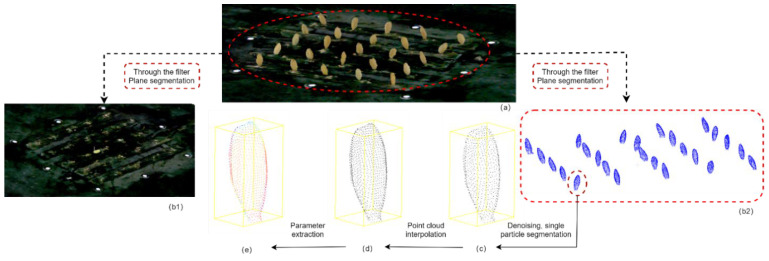
Schematic diagram of grain segmentation and data enhancement, (**a**) Original point cloud data, (**b1**) Sample plane point cloud data, (**b2**) Original rice grains point cloud data, (**c**) Single particle rice grain point cloud data, (**d**) Rice grain point cloud data with up-sampling, (**e**) Rice grain point cloud data with normal vector fusion.

**Figure 4 sensors-23-06331-f004:**
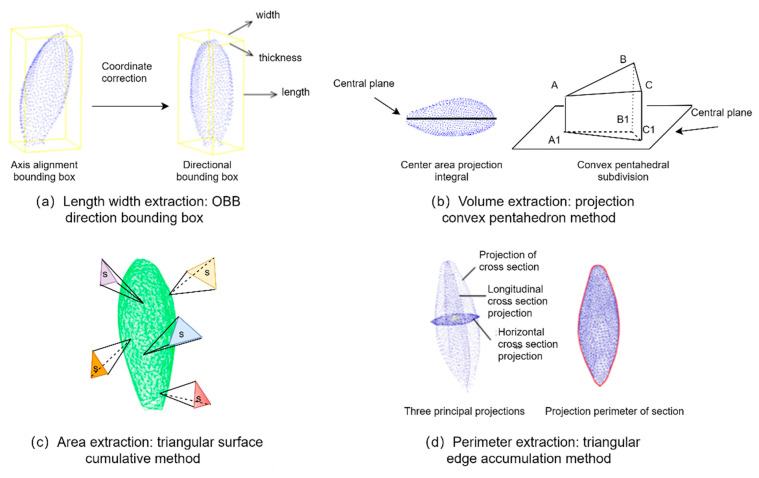
Schematic diagram of grain parameter extraction based on 3D point clouds.

**Figure 5 sensors-23-06331-f005:**
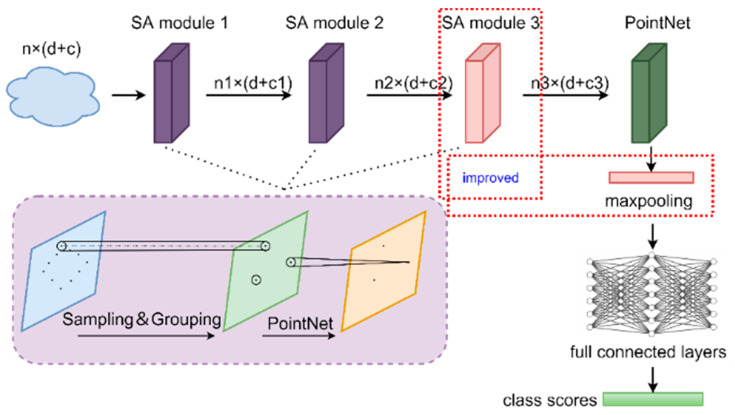
Improved PointNet++ network structure diagram. The red rectangles indicated the improved modules.

**Figure 6 sensors-23-06331-f006:**
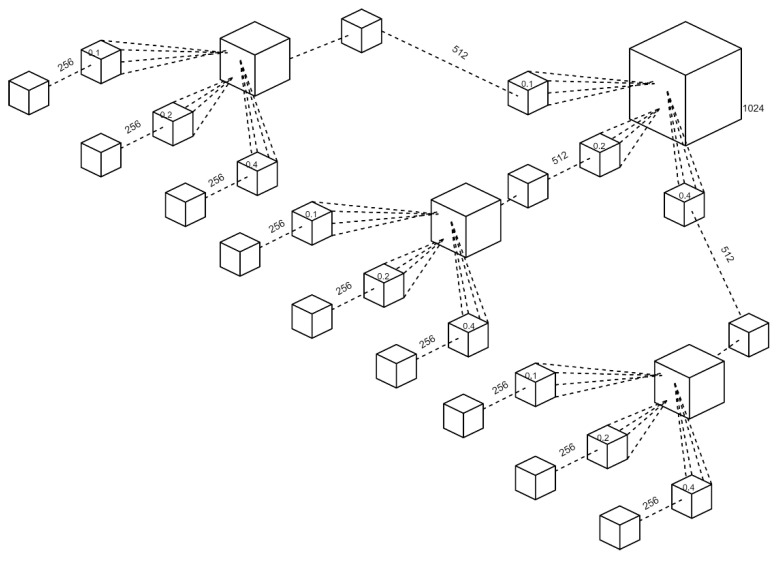
The sampling and grouping for set abstraction module.

**Figure 7 sensors-23-06331-f007:**
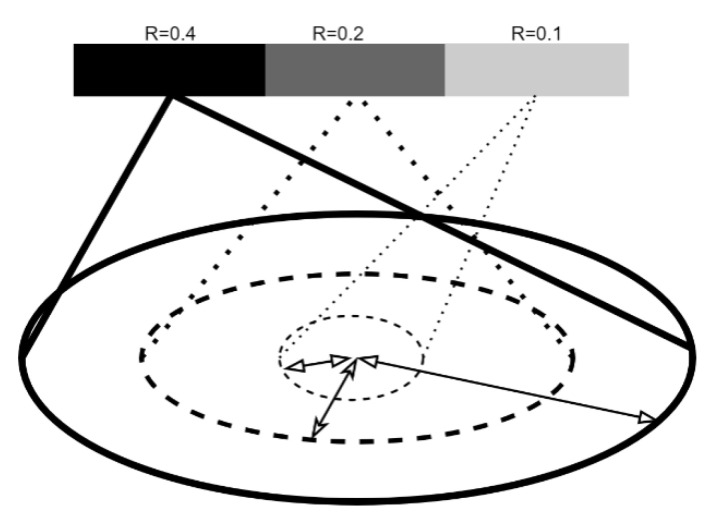
Multi-scale sampling module for rice grain point cloud grouping in radius of 0.1, 0.2 and 0.4 mm.

**Figure 8 sensors-23-06331-f008:**
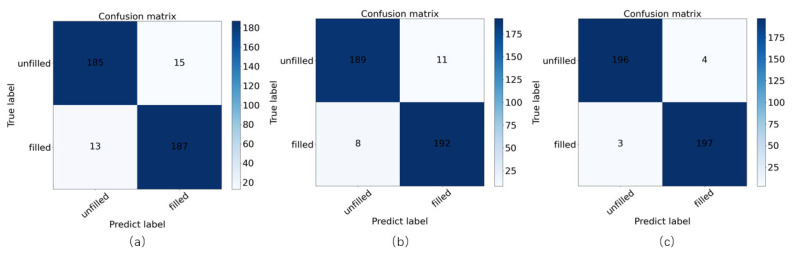
Confusion matrix of filled/unfilled rice grains of Improved PointNet++, (**a**) PointNet++ without data augmentation, (**b**) Improved PointNet++ without data augmentation, (**c**) Improved PointNet++ with data augmentation.

**Figure 9 sensors-23-06331-f009:**
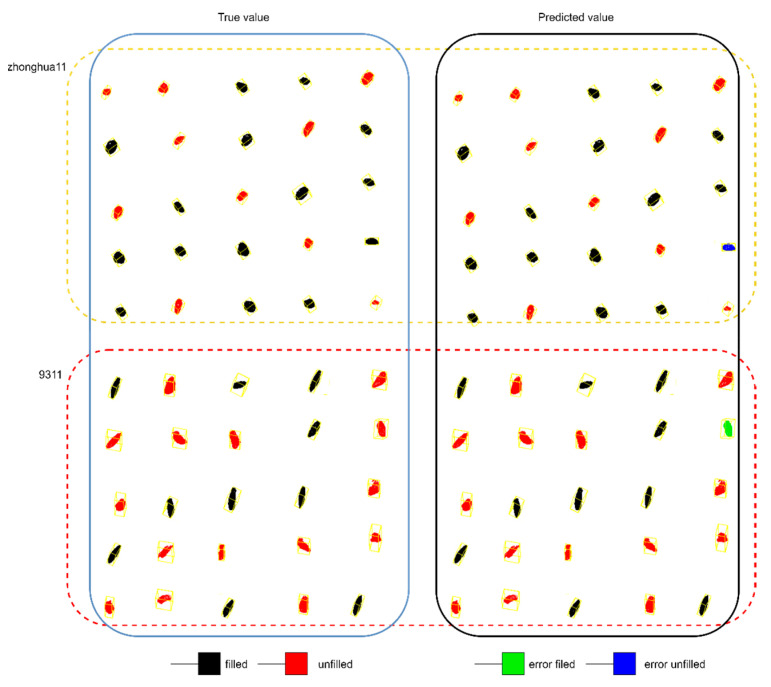
The filled/unfilled rice grain visual prediction for Zhonghua 11 and 9311 rice varieties.

**Table 1 sensors-23-06331-t001:** The parameters of the ReeyeePro scanner.

The Equipment Parameters	Parameter
The light source	white
Precision	0.05 mm
The scanning area	210 × 150 mm
Effective distance	290–490 mm
Maximum scan range	200 × 200 × 200 mm

**Table 2 sensors-23-06331-t002:** The parameters of the rice grains.

Parameter	Abbreviation	Parameter	Abbreviation
Length	L	Projected Perimeter in x direction	PPx
Width	W	Projected Area in y direction	PAx
Thickness	H	Projected Perimeter in y direction	PPx
Volume	V	Projected Area in z direction	PAx
Surface	S	Projected Perimeter in z direction	PPx
Projected Area in x direction	PAx		

**Table 3 sensors-23-06331-t003:** Grain classification results based on different deep learning models.

Classification Target	Model	Up-Sampling and Normal Vector Fusion	Accuracy	F1 Score	Efficiency
Filled/unfilled	PointNet	-	87.00%	86.00%	7.65 s
	PointNet	√	93.75%	92.78%	7.98 s
	PointConv	-	88.25%	86.60%	12.44 s
	PointConv	√	92.25%	91.67%	14.43 s
	PointNet++	-	91.75%	92.00%	9.75 s
	PointNet++	√	97.25%	91.80%	13.62 s
	Improved PointNet++	-	92.96%	95.28%	12.75 s
	Improved PointNet++	√	98.50%	98.25%	16.29 s

√: Data augmentation with up-sampling and normal vector fusion.

**Table 4 sensors-23-06331-t004:** PointNet++ parameter ablation experiment.

Up-Sampling	Normal Vector Fusion	Model	Accuracy
-	-	Improved PointNet++	92.96%
√	-	Improved PointNet++	94.50%
-	√	Improved PointNet++	97.75%
√	√	Improved PointNet++	98.50%

√: Using corresponding data augmentation methods.

**Table 5 sensors-23-06331-t005:** Grain classification results of machine learning model.

Classification Target	Method	Accuracy	F1 Score
Filled/unfilled	CART	89.94%	89.79%
	RF	90.08%	89.88%
	SVM	90.96%	90.84%
	NB	88.96%	88.34%
	BP	91.06%	90.59%
	XGBoost	91.99%	91.75%

## Data Availability

The raw datasets generated and used in this study are available are available upon request by contacting the first or corresponding authors.

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
