# Peer review of "A Novel Method for Filled/Unfilled Grain Classification Based on Structured Light Imaging and Improved PointNet++"

_sensors, 2023, doi:10.3390/s23146331_

Round 1

Reviewer 1 Report

First of all, I would like to congratulate the authors for such an interesting article proposed and of great interest for agriculture, looking for a higher yield in rice cultivation by discarding the empty grain.

Even so, I would like to make a series of observations after analyzing this work:

- Initially it is indicated that the methodology currently used for the detection of this empty grain is by air, but there is no brief description of this method.

- When implementing the new 3D technology, it is well focused and explained, but the cost, both financial and in terms of time, of carrying out this technique and comparing it with the one currently used is not detailed.

Overall, this methodology appears to be well designed to address the goal of identifying and classifying full and empty rice grains, but I think the article supports a comparison between the existing methodology and this innovative one.

With such a comparison, we would have a clearer idea of whether this new technique is an asset for improvement in the rice sector.

It is observed that the bibliographical references could also be increased by making the aforementioned comparison.

This study could be expanded a little more, since it is a bit incomplete since it does not make a comparison between the current methodology and the new one explained.

A comparative analysis with other machine learning algorithms has been carried out, which provides a fair and objective evaluation of the effectiveness of the proposed method. Overall, the study is valuable to the scientific community and may have practical applications in the food and agricultural industry.

Note the general improvement proposed to introduce the comparison with the air-grain classification method.

First of all, I would like to congratulate the authors for such an interesting article proposed and of great interest for agriculture, looking for a higher yield in rice cultivation by discarding the empty grain.

Even so, I would like to make a series of observations after analyzing this work:

- Initially it is indicated that the methodology currently used for the detection of this empty grain is by air, but there is no brief description of this method.

- When implementing the new 3D technology, it is well focused and explained, but the cost, both financial and in terms of time, of carrying out this technique and comparing it with the one currently used is not detailed.

Overall, this methodology appears to be well designed to address the goal of identifying and classifying full and empty rice grains, but I think the article supports a comparison between the existing methodology and this innovative one.

With such a comparison, we would have a clearer idea of ​​whether this new technique is an asset for improvement in the rice sector.

It is observed that the bibliographical references could also be increased by making the aforementioned comparison.

This study could be expanded a little more, since it is a bit incomplete since it does not make a comparison between the current methodology and the new one explained.

A comparative analysis with other machine learning algorithms has been carried out, which provides a fair and objective evaluation of the effectiveness of the proposed method. Overall, the study is valuable to the scientific community and may have practical applications in the food and agricultural industry.

Note the general improvement proposed to introduce the comparison with the air-grain classification method.

Author Response

To Reviewer 1 Comments:

First of all, I would like to congratulate the authors for such an interesting article proposed and of great interest for agriculture, looking for a higher yield in rice cultivation by discarding the empty grain. Even so, I would like to make a series of observations after analyzing this work.

Response: Thanks for the reviewer’s positive comments. In this study, we developed a novel method based on structured light imaging and improved PointNet++ to achieve accurate filled/unfilled rice grain identification, which was of great significance for rice breeding and genetic analysis. According to the suggestions, we have carefully revised our manuscript, and all the concerns have addressed point-by-point.

  1. Initially it is indicated that the methodology currently used for the detection of this empty grain is by air, but there is no brief description of this method.

Response:Thanks for the reviewer’s suggestion. The brief description of the water sedimentation or air separation methods have been supplemented as follows “Water sedimentation and air separation method utilize the internal cavity of the unfilled grains to achieve the separation of the filled/unfilled grain by adjusting external forces.”(Lines 43-45)

  1. When implementing the new 3D technology, it is well focused and explained, but the cost, both financial and in terms of time, of carrying out this technique and comparing it with the one currently used is not detailed.

Response:Thanks for the reviewer’s suggestions. Compared with binocular vision Iterative reconstruction technology, Structured light Iterative reconstruction technology has lower time cost, which is mainly generated by camera calibration, point cloud reconstruction, and point cloud post-processing. And the detailed comparison was added as follows:

“As for the binocular stereo vision, motion structure, and space carving methods, camera parameter calibration should be required for each shot, while the obtained point clouds were sparse and time-consuming for rice grain 3D reconstruction [15]. While the 3D structured light imaging method would automatically obtain high-density rice grain point clouds with texture information, avoiding the impact of environmental noise.”(Lines 54-59)

  1. Overall, this methodology appears to be well designed to address the goal of identifying and classifying full and empty rice grains, but I think the article supports a comparison between the existing methodology and this innovative one. With such a comparison, we would have a clearer idea of whether this new technique is an asset for improvement in the rice sector.

Response:Thanks for the reviewer’s positive comments.

  1. It is observed that the bibliographical references could also be increased by making the aforementioned comparison.

Response:Thanks for the reviewer’s comments. The reference was increased as follows:

“Ping, X. X.; Liu, Y.; Dong, X. M.; Zhao, Y. Q.; Yan, Z. 3-D reconstruction of textureless and high-reflective target by polarization and binocular stereo vision. Journal Of Infrared and Millimeter Waves. 2017, 36(4), 432-438.”(Reference 15)

  1. This study could be expanded a little more, since it is a bit incomplete since it does not make a comparison between the current methodology and the new one explained. A comparative analysis with other machine learning algorithms has been carried out, which provides a fair and objective evaluation of the effectiveness of the proposed method. Overall, the study is valuable to the scientific community and may have practical applications in the food and agricultural industry.

Response:Thanks for the reviewer’s positive comments. The comparison between the current methodology and our method was added as follows, “As for the binocular stereo vision, motion structure, and space carving methods, camera parameter calibration should be required for each shot, while the obtained point clouds were sparse and time-consuming for rice grain 3D reconstruction [15]. While the 3D structured light imaging method would automatically obtain high-density rice grain point clouds with texture information, avoiding the impact of environmental noise.”(Lines 54-59)

Reviewer 2 Report

The paper is focused on testing machine learning methods for rice grain classification.
Although experiment setup is clear, the following questions must be addressed by the authors:
1. The authors need to explain more precisely how the experimental rice grains were selected. Did the rice grains grow in the same region or various regions? Why the filled and unfilled grains were taken in 50/50 proportion - it is not a random selection as it is stated in the work. To say it in a more general way, how can we be sure that the training sample represent population and what population is?
2. What are the main reasons for using a certain hardware (ReeyeePro scanner) for experimental work, and can it be used for industrial purposes for grain classification? In other words, can this experiment be scaled?
3. When evaluating the models, it is important to see not just values of metrics, but confusion matrices as well with analysis of user's/producer's accuracies, omission and commission errors. I have a feeling that these errors have different levels of importance in grain selection procedures.

The overall impression is that the work is rather technical and the depth of results is insufficient. In particular, it is not clear how far the proposed experiments from real-life implementation.

The language is generally acceptable, however there are some confusing points: "Error! Reference source not found.." (pp. 2, 6, 12)

Author Response

To Reviewer 2 Comments:

The paper is focused on testing machine learning methods for rice grain classification. Although experiment setup is clear, the following questions must be addressed by the authors:

Response: Thanks for the reviewer’s positive comments. According to the suggestions, we have carefully revised our manuscript, and all the concerns have addressed point-by-point as follows.

  1. The authors need to explain more precisely how the experimental rice grains were selected. Did the rice grains grow in the same region or various regions? Why the filled and unfilled grains were taken in 50/50 proportion - it is not a random selection as it is stated in the work. To say it in a more general way, how can we be sure that the training sample represent population and what population is?

Response: Thanks for the reviewer’s comments. “In this study, the experimental materials were selected from the japonica and indica rice varieties Zhonghua 11 and 9311, which belonged to japonica and indica rice sub-species respectively, in which the rice grains were randomly selected for the training set, so the training samples were able to acquire representative morphological characteristics of rice grains. Each rice variety comprised of 500 filled rice grains and 500 unfilled rice grains to ensure sample balance, resulting in a total of 1000 rice grains. The rice grains were randomly selected, while the filled and unfilled rice grains were identified by the air separation.” (Lines 87-93)

  1. What are the main reasons for using a certain hardware (ReeyeePro scanner) for experimental work, and can it be used for industrial purposes for grain classification? In other words, can this experiment be scaled?

Response: Thanks for the reviewer’s comments. The ReeyeePro scanner was an economic and portable 3D imaging device, which cost about 20s to obtain complete rice grain point clouds. If large-scale industrial experiments were carried out, more scanners should be employed and automatic analysis software should be improved, while this study would provide a novel and effective method for filled/unfilled grain recognition.

  1. When evaluating the models, it is important to see not just values of metrics, but confusion matrices as well with analysis of user's/producer's accuracies, omission and commission errors. I have a feeling that these errors have different levels of importance in grain selection procedures.

Response: Thanks for the reviewer’s comments. The classification result for the confusion matrix of filled/unfilled rice grains, was shown as Figure 8. As for the Improved PointNet++ model, the results proved that the misjudgments of filled and unfilled grains were 3 and 4, respectively, and we agreed that these errors have different levels of importance in grain selection procedures. (Lines 298-301)

  1. The language is generally acceptable, however there are some confusing points: "Error! Reference source not found." (pp. 2, 6, 12)

Response: Thanks for the comments. We have checked and corrected the references. (pp. 2, 6, 12).

Reviewer 3 Report

The paper proposes a novel method for filled/unfilled grain classification in rice based on structured light imaging and an improved PointNet++ network, demonstrating its potential for enhancing rice breeding and genetic analysis.

Although the approach is relevant and the contribution interesting, it is crucial to address and clarify some significant questions :

1- How does the removal or addition of point clouds in order to maintain a consistent number affect the overall performance and accuracy of the classification results?

2- What is the computational cost or efficiency of the improved PointNet++ network compared to other existing methods for filled/unfilled grain classification?

3- How does the proposed method perform when applied to different types or varieties of rice grains, considering potential variations in size, shape, and texture?

4- Can the findings from this study be extended to other agricultural or food products that require similar classification tasks based on point cloud data?

Author Response

To Reviewer 3 Comments:

The paper proposes a novel method for filled/unfilled grain classification in rice based on structured light imaging and an improved PointNet++ network, demonstrating its potential for enhancing rice breeding and genetic analysis. Although the approach is relevant and the contribution interesting, it is crucial to address and clarify some significant questions:

Response: Thanks for the reviewer’s positive comments. According to the suggestions, we have carefully revised our manuscript, and all the questions have addressed and clarified point-by-point as follows.

  1. How does the removal or addition of point clouds in order to maintain a consistent number affect the overall performance and accuracy of the classification results?

Response: Thanks for the reviewer’s comments. The removal or addition of point clouds were conducted to acquire the same number of point clouds, which was necessary for the PointNet++ model implementation. In the conclusion, “The data augmentation techniques including up-samping and normal vector fusion were adopted to increase point cloud density and enrich point cloud features, which was proved to be of great significance for the filled/unfilled grain classification improvement. And the results proved that the data augmentation could promote the classification accuracy by 5.54% for the Improved PointNet++.”(Lines 341-345)

  1. What is the computational cost or efficiency of the improved PointNet++ network compared to other existing methods for filled/unfilled grain classification?

Response: Thanks for the reviewer’s comments. “The results demonstrated that the data augmentation would dramatically increase the classification accuracy by over 4.00%, but only decrease the efficiency by average 2.43s. The data augmentation would increase the classification accuracy by 5.54% for the Improved PointNet++, which was proved to be an effective method for filled/unfilled grain classification improvement, as was shown in Table 3.” (Lines 272-276)

  1. How does the proposed method perform when applied to different types or varieties of rice grains, considering potential variations in size, shape, and texture?

Response: Thanks for the reviewer’s comments. This study randomly selected rice grains from 9311 and Zhonghua 11, which belonged to japonica and indica rice sub-species, respectively. The indica had a longer grain shape, with a length of more than three times the width, flat, short and thin fur, generally without awns. However, the japonica had a larger and shorter grain shape, with a length of about twice the width. Therefore, the training set had representative morphological characteristics of rice grains, which had a large scale range for grain length, width and height, volume surface area, projected area and projected perimeter. Therefore, this method was feasible for applying to different types or varieties of rice grains.

  1. Can the findings from this study be extended to other agricultural or food products that require similar classification tasks based on point cloud data?

Response: Thanks for the reviewer’s comments. This study had demonstrated a novel and effective method for filled/unfilled grain recognition, and the findings of this study can be extended to other agricultural products or foods that require similar classification.

Round 2

Reviewer 1 Report

After the improvements have been made, the document can be presented as finished.

Author Response

Thanks for the reviewer’s positive comment

Reviewer 2 Report

Thanks for adressing some of my points. Nevertheless, I would to emphasize my question raised in the first review round again: how can we be sure that the training sample represent population and what population is?

To make it clearer, you state:
the experimental materials were selected from the japonica and indica rice varieties Zhonghua 11 and 9311

My question is: do you pretend to use your method only for detecting unfilled grains selected out of the mentioned rice varieties? If not, how can you be sure that your machine learning methods still work well?
Another question: does 50/50 proportion of filled/unfilled grains represent what you usually encounter when dealing with rice crops? If the real proportion is different, say 15-20% of unfilled grains, then is it plausible to take such a propotion in the training set.

Author Response

To Reviewer 2 Comments:

Thanks for adressing some of my points. Nevertheless, I would to emphasize my question raised in the first review round again: how can we be sure that the training sample represent population and what population is?

Response: Thanks for the reviewer’s comment. In this study, we are trying to demonstrate a novel method for filled/unfilled grain classification based on structured light imaging and Improved PointNet++. Although the training samples were only selected from two rice varieties Zhonghua 11 and 9311, they belonged to japonica and indica rice sub-species, and we believed that the training samples could represent the general japonica and indica rice varieties. The results proved that the classification accuracy of Improved PointNet++ was 98.50% outperformed the PointNet 93.75%, PointConv 92.25%, and the optimal machine learning model 91.99%, which were conducted in the same samples.

1.1To make it clearer, you state:the experimental materials were selected from the japonica and indica rice varieties Zhonghua 11 and 9311. My question is: do you pretend to use your method only for detecting unfilled grains selected out of the mentioned rice varieties? If not, how can you be sure that your machine learning methods still work well?

Response: Thanks for the question again. Firstly, the Zhonghua 11 and 9311 are representative rice varieties of japonica and indica, in which the filled and unfilled grain were randomly selected, and the shape parameters had a wide range. Secondly, we are trying to demonstrate a novel method for filled/unfilled grain classification based on structured light imaging and Improved PointNet++, which outperformed PointNet, PointConv, and the machine learning model based on the same samples. Finally, in order to verify generalization ability of this method, 25 grains of Fengliangyou 4 and Guangliangyouxiang 66 were tested based on the Improved PointNet++ model, and the classification accuracy was 95.83% and 98.00% respectively (Lines 303-305), which could be optimized by adding training samples of the new rice varieties.

1.2Another question: does 50/50 proportion of filled/unfilled grains represent what you usually encounter when dealing with rice crops? If the real proportion is different, say 15-20% of unfilled grains, then is it plausible to take such a propotion in the training set.

Response: Thanks for the reviewer’s suggestion. The reason for 50/50 proportion of filled/unfilled grains was taken in the study, was to ensure sample balance and prevent classification bias during the model training, which was also a routine operation in the deep learning methods. In addition, the dynamic range of rice grain filling rate was very large for different varieties, so the specific proportion for a rice variety might be not suitable for other rice varieties.